# Geometric Structure of PINN Latent Space for Burger's Equation: Low-Dimensional Manifolds and Initial Condition Encoding

## Abstract

Understanding how Physics-Informed Neural Networks (PINNs) encode complex physical systems and the influence of parameters like initial conditions within their latent representations is crucial for interpretability and application. This study investigates the geometric structure of the 10-dimensional latent space generated by a PINN solving the 2D Burger's equation across 25 different initial conditions. Using Principal Component Analysis and subspace similarity measures, we analyze the set of latent vectors for each initial condition as a potential low-dimensional manifold embedded in $\mathbb{R}^{10}$, comparing and contrasting these structures across the dataset of simulated solutions. The analysis reveals a highly organized latent space; globally, the latent vectors occupy an effectively 6-dimensional subspace capturing over 99% of variance. For each individual initial condition, the latent vectors form a distinct, approximately 3-dimensional affine manifold, a structure remarkably consistent across all tested conditions. Crucially, the primary effect of changing the initial condition is encoded as a translation of this 3D manifold along a nearly one-dimensional path within the 10-dimensional latent space, strongly aligned with the global principal component. Furthermore, these 3D manifolds are remarkably parallel to each other, exhibiting an average subspace similarity exceeding 0.98, with only subtle, low-dimensional variations in their orientation. These findings demonstrate that the PINN learns a highly structured and efficient parameterization where initial conditions select specific, geometrically simple, and highly related low-dimensional structures within the overall latent space, offering valuable insights into the network's internal encoding mechanisms and suggesting potential avenues for model interpretation and compression.[1]

## 1 Introduction

Physics-Informed Neural Networks (PINNs) represent a significant advancement in solving partial differential equations (PDEs) by embedding the governing physical laws directly into the neural network architecture and training objective.

This approach offers compelling advantages, such as the ability to handle complex geometries and scenarios with limited observational data, providing a mesh-free alternative to traditional numerical techniques. However, despite their successes, PINNs, like many deep learning models, often function

---

[1]This paper, including the idea and the research analysis, was fully generated and written by Denario, a multi-AI agent system. All the input and output files, together with the original paper, can be found in the supplementary material. The Denario code is available in the supplementary material and a YouTube video demonstrating the end-to-end research pipeline with Denario is available in the anonymised YouTube channel at this link.

Submitted to 1st Open Conference on AI Agents for Science (agents4science 2025). Do not distribute.

as "black boxes," obscuring the precise mechanisms by which they learn and represent the underlying physical phenomena. Understanding how these networks encode complex solution landscapes and incorporate the influence of problem parameters, such as initial and boundary conditions, is paramount for enhancing their reliability, interpretability, and facilitating downstream applications like model compression or transfer learning.

A central element within many neural network architectures, including PINNs, is the latent space. This intermediate representation layer compresses high-dimensional input data into a more abstract, often lower-dimensional, form. In the context of a PINN solving a PDE, the latent space typically holds a learned encoding of the physical state of the system at specific points in space and time $(x, t)$.

Investigating the structure of this latent space provides a window into how the network perceives and processes the physics. A fundamental challenge lies in deciphering how the latent representation varies across the physical domain $(x, t)$ and, critically, how this variation changes in response to modifications in the problem's parameters, such as the initial condition. The difficulty is compounded by the potentially high dimensionality of the latent space (10 dimensions in this study) and the unknown, potentially complex non-linear geometric structures formed by the collection of latent vectors corresponding to a given physical solution. For a specific initial condition, the set of latent vectors $\{L(x, t)\}$ sampled over a grid of $(x, t)$ points forms a point cloud in this 10-dimensional space, whose intrinsic structure and relationship to other such point clouds generated by different initial conditions are not *a priori* understood.

This study focuses on dissecting the geometric structure of the 10-dimensional latent space generated by a PINN trained to solve the 2D Burger's equation. The 2D Burger's equation is a canonical non-linear PDE widely used as a simplified model for complex fluid dynamics phenomena like turbulence and shock formation, known for its rich dynamic behavior highly sensitive to initial conditions. We specifically examine how the PINN's latent representation of the solution changes across 25 distinct initial conditions. For each initial condition, we treat the collection of latent vectors $\{L(x, t)\}$ sampled across a discrete grid of $(x, t)$ points as a dataset forming a point cloud in $\mathbb{R}^{10}$. Our primary objective is to analyze the geometric properties of these point clouds, characterizing their effective dimensionality, shape, and how these characteristics compare and contrast across the ensemble of 25 initial conditions. We hypothesize that despite the complexity of the Burger's equation and the high dimensionality of the latent space, the network may learn a structured and perhaps simple encoding where the latent point clouds exhibit low-dimensional geometric properties and are related across initial conditions by simple transformations.

To achieve this, we employ a suite of geometric analysis techniques. Principal Component Analysis (PCA) is utilized extensively to quantify the dominant directions of variation and determine the effective low dimensionality of the latent vector point clouds, both for the global collection of all latent vectors across all initial conditions, and for the point cloud corresponding to each individual initial condition. Furthermore, we employ subspace similarity measures to quantitatively compare the orientations of the principal subspaces learned for different initial conditions. By systematically analyzing the centroids of these point clouds and the relationship between their principal components and the global latent space structure, we aim to build a comprehensive picture of how the PINN encodes the effect of varying initial conditions within its learned representation. This approach allows us to test whether changes in initial conditions correspond to simple, predictable geometric transformations, such as translations or rotations, of a fundamental latent structure.

Our analysis reveals a highly structured organization within the latent space. We find that, while the latent space is 10-dimensional, the entire collection of latent vectors across all initial conditions occupies an effectively 6-dimensional subspace, capturing over 99% of the total variance.

Strikingly, for each individual initial condition, the corresponding set of latent vectors forms a distinct, approximately 3-dimensional affine manifold. This 3D structure is remarkably consistent in its intrinsic dimensionality and variance distribution across all 25 tested initial conditions. Crucially, the primary effect of changing the initial condition is encoded as a translation of this consistent 3D manifold. These manifold centroids trace a nearly one-dimensional path within the 10-dimensional latent space, strongly aligned with the dominant global principal component. Moreover, the orientations of these 3D manifolds are exceptionally similar, exhibiting an average subspace similarity exceeding 0.98, indicating they are nearly parallel with only subtle, low-dimensional variations in their alignment. These findings demonstrate that the PINN learns a highly efficient and structured parameterization where initial conditions select specific, geometrically simple, and highly related

low-dimensional structures within the overall latent space, offering valuable insights into the network's internal encoding mechanisms and suggesting potential avenues for model interpretation and compression.

# 2 Methods

The objective of this study is to dissect the geometric structure of the 10-dimensional latent space generated by a Physics-Informed Neural Network (PINN) trained to solve the 2D Burger's equation. We investigate how the latent representations corresponding to different initial conditions are organized within this space and how their structure relates across an ensemble of 25 distinct initial conditions. Our methodology involves data preparation, applying Principal Component Analysis (PCA) to characterize the dimensionality and variance distribution of latent vector sets, and employing subspace similarity measures to compare the orientations of principal subspaces across different initial conditions.

## 2.1 Latent Space Data Preparation

The data used in this analysis originates from a pre-trained PINN solving the 2D Burger's equation over a specified spatiotemporal domain. The data was provided as a NumPy array `data_bundle` with dimensions $(101, 103, 25, 13)$. These dimensions correspond to spatial grid points ($x$-coordinate), time steps ($t$), initial condition index, and features, respectively. The spatial grid consists of 101 points along the $x$-axis, and the temporal domain is discretized into 103 time steps. The dataset includes solutions and latent space representations for 25 different initial conditions. The features dimension (size 13) contains the predicted solution components (e.g., velocity fields $u$ and $v$) and the 10-dimensional latent vector output by an intermediate layer of the PINN for each spatial point ($x$) and time step ($t$) under a specific initial condition.

The 10-dimensional latent space data was extracted from the last 10 components of the features dimension. This resulted in a tensor `latent_space_data` with dimensions $(101, 103, 25, 10)$. Each element `latent_space_data[i, j, k, :]` represents the 10-dimensional latent vector $L(x_i, t_j, \text{IC}_k)$ corresponding to the spatial point $x_i$, time $t_j$, and the $k$-th initial condition $\text{IC}_k$. For each initial condition $k$, the set of latent vectors $\{L(x_i, t_j, \text{IC}_k)\}$ over all $i = 0..100$ and $j = 0..102$ forms a collection of $101 \times 103 = 10403$ points in the 10-dimensional latent space $\mathbb{R}^{10}$. This collection is treated as a point cloud representing the PINN's latent encoding of the physical solution for initial condition $\text{IC}_k$.

## 2.2 Geometric Analysis Techniques

To analyze the structure of these point clouds and their relationships, we employed Principal Component Analysis (PCA) [4, 9, 6] and subspace similarity measures.

### 2.2.1 Principal Component Analysis (PCA)

PCA is a statistical procedure that uses an orthogonal transformation to convert a set of observations of possibly correlated variables into a set of values of linearly uncorrelated variables called principal components [4, 5]. This transformation is defined in such a way that the first principal component has the largest possible variance (that is, accounts for as much of the variability in the data as possible), and each succeeding component in turn has the highest variance possible under the constraint that it is orthogonal to the preceding components [4]. The principal components are the eigenvectors of the data's covariance matrix, and their corresponding eigenvalues represent the variance along those directions [4, 5].

In this study, PCA was applied in several contexts:

- **Global PCA:** PCA was applied to the entire collection of latent vectors across all spatial points, time steps, and initial conditions. The `latent_space_data` tensor was reshaped into a $2D$ matrix of size $(101 \times 103 \times 25, 10)$, effectively treating all $10403 \times 25 = 260075$ latent vectors as a single dataset in $\mathbb{R}^{10}$. This global PCA reveals the overall dimensionality and dominant directions of variation within the latent space spanned by all observed states. The eigenvalues were used to calculate the percentage of total variance explained by each

principal component and the cumulative variance, providing an estimate of the effective global dimensionality.

- **Per-Initial Condition PCA:** For each of the 25 initial conditions, PCA was applied independently to the set of $10403$ latent vectors $\{L(x_i, t_j, \text{IC}_k)\}$ corresponding to that specific initial condition $k$. For each IC $k$, the data `latent_space_data[:, :, k, :]` was reshaped into a $2D$ matrix of size $(10403, 10)$. This per-IC PCA characterizes the intrinsic dimensionality and shape of the point cloud associated with a single physical solution. The centroid (mean vector) $C_k$ of the point cloud for IC $k$ was calculated, and the eigenvalues and eigenvectors (principal components) of its covariance matrix were obtained. The eigenvalues indicate the variance along the principal directions, and the eigenvectors form an orthonormal basis for the principal subspace capturing the data's variation. The cumulative variance explained by the principal components for each IC was analyzed to determine the effective intrinsic dimensionality of the manifold for that specific initial condition.

- **PCA on Centroids:** The centroids $C_k$ for each of the 25 initial conditions are 10-dimensional vectors. These 25 centroid vectors were collected into a $2D$ matrix of size $(25, 10)$. PCA was applied to this matrix to analyze the geometric arrangement of the manifold centroids in the latent space. This reveals whether the variation in initial conditions primarily translates the latent manifold along a low-dimensional path or occupies a more complex structure in the latent space.

For all PCA applications, the data was centered by subtracting the mean before computing the covariance matrix and performing the eigenvalue decomposition.

### 2.2.2 Subspace Similarity Measures

To compare the orientations of the principal subspaces identified by the per-IC PCA, we employed subspace similarity measures. For each initial condition $k$, the per-IC PCA yields a set of principal components $\{v_{k,1}, v_{k,2}, \ldots, v_{k,10}\}$ ordered by their corresponding eigenvalues. Based on the cumulative variance explained, we determined an effective intrinsic dimensionality $d_{ic}$ for the individual manifolds (e.g., the number of components capturing 95% of variance) [4, 6]. The principal subspace for IC $k$ is then approximated by the span of its first $d_{ic}$ principal components, $\text{span}\{v_{k,1}, \ldots, v_{k,d_{ic}}\}$.

To quantify the similarity between the principal subspaces of two initial conditions $k$ and $j$, we compared their sets of principal vectors $\{v_{k,1}, \ldots, v_{k,d_{ic}}\}$ and $\{v_{j,1}, \ldots, v_{j,d_{ic}}\}$. A quantitative measure of subspace similarity is given by the principal angles between the two subspaces. Alternatively, for small $d_{ic}$, the similarity can be approximated by comparing corresponding principal vectors. For instance, the alignment of the primary direction of variation is measured by the absolute dot product $|v_{k,1} \cdot v_{j,1}|$. A value close to 1 indicates strong alignment, while a value close to 0 indicates orthogonality. We computed these measures for pairs of corresponding principal vectors (e.g., $v_{k,1}$ vs $v_{j,1}$, $v_{k,2}$ vs $v_{j,2}$) across all pairs of initial conditions to assess the consistency in manifold orientation.

A high average subspace similarity across all pairs of ICs indicates that the principal directions of variation for the latent manifolds are largely parallel, implying that the manifolds are primarily translated versions of each other.

### 2.3 Analysis Workflow

The analysis was structured in a sequence of steps to progressively reveal the geometric structure of the latent space [8, 7] and the encoding of initial conditions:

### 2.3.1 Initial Exploratory Data Analysis

We began by performing global PCA on the entire collection of latent vectors to understand the overall distribution and effective dimensionality of the combined dataset [4, 6]. Concurrently, we performed per-IC PCA for each of the 25 initial conditions to obtain individual centroids and principal components, characterizing the typical intrinsic dimensionality and variance structure of a single manifold [3]. Finally, PCA was applied to the set of 25 centroids to understand how the mean positions of the manifolds are organized [6].

### 2.3.2 Characterization of Individual Manifolds

Based on the per-IC PCA results, we determined the effective intrinsic dimensionality $d_{ic}$ for the latent point cloud of each initial condition. We approximated each point cloud as an affine subspace defined by its centroid $C_k$ and the span of its first $d_{ic}$ principal component vectors $V_k = [v_{k,1}, \ldots, v_{k,d_{ic}}]$ [4, 3, 6]. The eigenvalues associated with these vectors provided insight into the extent of the manifold along each principal direction [4, 6].

### 2.3.3 Comparative Analysis Across Initial Conditions

We systematically compared the characterized manifolds across the 25 initial conditions. The analysis of centroids (PCA on $\{C_k\}$) revealed the structure of the path traced by the manifold centers as the initial condition changes [6]. Subspace similarity measures were computed for pairs of principal subspaces span($V_k$) to quantify how similarly oriented the manifolds are [4, 6]. By combining the information from centroid locations and manifold orientations, we assessed whether the primary effect of changing the initial condition is a simple translation, a rotation, or a more complex transformation of a fundamental latent structure [6]. We also specifically analyzed the set of first principal vectors $\{v_{k,1}\}$ across all ICs using PCA to see if the dominant direction of variation for individual manifolds exhibits a structured, possibly low-dimensional, variation across ICs [6].

### 2.3.4 Relation to Global Latent Space Structure

Finally, we related the local structures (individual manifolds) to the global structure identified by the global PCA. We projected the centered latent vectors $(L_k - C_k)$ for each IC $k$ onto the dominant subspace identified by the global PCA to see how much of the per-IC variance is aligned with the global principal directions. We also examined the alignment between the per-IC principal subspaces span($V_k$) and the global principal subspace span($U_{glob}$).

### 2.3.5 Synthesis

The findings from these analyses were synthesized to provide a comprehensive geometric description of the PINN's latent space [1]. We described the typical intrinsic dimensionality of the latent representation for a single solution, the extent to which these representations form affine manifolds, how these manifolds are related across different initial conditions (e.g., by translation along a low-dimensional path, by consistent orientation), and how these local structures relate to the overall structure of the latent space. This synthesis allowed us to draw conclusions about how the PINN efficiently encodes the initial condition within its internal representation [2].

## 3 Results

The objective of this study was to investigate the geometric structure of the 10-dimensional latent space generated by a PINN solving the 2D Burger's equation, focusing on how different initial conditions are encoded within this space. Using Principal Component Analysis (PCA) and subspace similarity measures, we analyzed the latent vectors corresponding to 25 distinct initial conditions.

### 3.1 Global structure of the latent space

We began by analyzing the overall structure of the latent space by performing PCA on the entire collection of latent vectors generated across all spatial points, time steps, and the 25 initial conditions. This global analysis, as described in the Methods, treats all $101 \times 103 \times 25 = 260075$ latent vectors as a single dataset in $\mathbb{R}^{10}$. The variance explained by each principal component (PC) provides insight into the intrinsic dimensionality and dominant directions of variation within the aggregated latent representation.

The results of this global PCA reveal a significant concentration of variance in the leading principal components. As shown in the scree plot in Figure 1, the first principal component (PC1) alone captures 60.12% of the total variance. The second (PC2) and third (PC3) components capture an additional 23.44% and 12.93%, respectively. Cumulatively, the first three global PCs account for 96.48% of the total variance. Including the fourth (1.30%), fifth (1.17%), and sixth (0.76%)

components brings the cumulative variance explained to 99.72%. The remaining four components individually explain less than 0.3% of the variance each.

This strong concentration of variance within the first six principal components demonstrates that the entire collection of latent vectors, despite residing in a 10-dimensional space, effectively occupies a much lower-dimensional subspace. The vast majority (>99%) of the variability observed in the latent representations across all tested physical states and initial conditions is captured by a 6-dimensional linear subspace. This suggests that the PINN learns an overall efficient encoding, where the complex dynamics across different conditions are constrained to a relatively low-dimensional manifold within the full latent space.

## 3.2  Intrinsic dimensionality of per initial condition manifolds

Next, we investigated the structure of the latent space corresponding to individual initial conditions. For each of the 25 initial conditions ($IC_k$, $k = 0, \ldots, 24$), we performed PCA independently on the $101 \times 103 = 10403$ latent vectors $\{L(x_i, t_j, IC_k)\}$ associated with that specific condition. This analysis aims to characterize the intrinsic dimensionality and shape of the latent point cloud representing the PINN's encoding of the solution for a fixed initial state.

The results show a remarkable consistency across all 25 initial conditions. As shown by the average scree plot in Figure 2 and the intrinsic dimensionality distribution in Figure 3, for every single IC, precisely 3 principal components were sufficient to explain over 95% of the variance within its corresponding latent point cloud. Quantitatively, the average cumulative variance explained by the first three per-IC principal components is 97.48%, with a very low standard deviation (0.15%). The average variance explained by the first, second, and third per-IC PCs were 59.61%, 23.72%, and 14.15%, respectively. The variance captured by the fourth per-IC PC and beyond drops sharply, with the average variance for the fourth PC being below 2%.

These findings strongly suggest that, for any given initial condition within the tested set, the PINN's latent representation of the spatiotemporal solution $\{L(x, t)\}$ forms an effectively 3-dimensional structure embedded in the 10-dimensional latent space. The high percentage of variance captured by the leading three components indicates that these structures are well-approximated by 3-dimensional affine manifolds (shifted linear subspaces), exhibiting limited non-linear deviations from this linear approximation within the scope of the tested conditions. This implies that the network has learned a consistent, low-dimensional basis for representing the state of the system over space and time for a fixed initial condition.

## 3.3  Geometric arrangement of manifold centroids

To understand how the latent representations differ across initial conditions, we analyzed the geometric arrangement of the centroids $C_k$ of the per-IC latent point clouds. Each centroid $C_k$ is a 10-dimensional vector representing the mean position of the latent manifold for initial condition $IC_k$. We collected these 25 centroid vectors and performed PCA on this ($25 \times 10$) matrix.

The results of this centroid PCA are striking, as shown in the scree plot in Figure 4. The first principal component of the centroids (CPC1) explains an overwhelming 99.86% of the total variance in the centroid positions. The second component (CPC2) explains only 0.10%, and the third (CPC3) explains 0.02%.

Projecting the centroids onto their principal components, as depicted in Figure 5 (2D projection) and Figure 6 (3D projection), reveals that they form an almost perfectly linear arrangement in the latent space. The centroids corresponding to initial conditions indexed 0 through 24 are ordered sequentially along this dominant, nearly one-dimensional direction defined by CPC1.

This finding is crucial: it indicates that the primary effect of changing the initial condition within this ensemble is to translate the entire 3D latent manifold corresponding to that condition along a specific, nearly one-dimensional path within the 10-dimensional latent space. This suggests that the PINN encodes the difference between initial conditions predominantly as a shift in the mean position of the learned solution manifold.

## 3.4 Comparison of manifold orientations

While the centroids reveal the translational differences between the manifolds, we also investigated whether the orientation or "shape" of the 3D manifolds changes across initial conditions. For each $IC_k$, the per-IC PCA yields a set of principal vectors $\{v_{k1}, v_{k2}, v_{k3}\}$ spanning the approximate 3D affine manifold. We compared these principal subspaces across different initial conditions.

We quantified the similarity between the 3-dimensional principal subspaces spanned by $\{v_{k1}, v_{k2}, v_{k3}\}$ for pairs of initial conditions $(IC_k, IC_l)$ using subspace similarity measures based on principal angles. The results, shown in the heatmap in Figure 7, indicate that the average subspace similarity score across all pairs of initial conditions was exceptionally high, measuring 0.986, with a standard deviation of only 0.014. The minimum observed similarity was 0.954. A similarity score close to 1 indicates that the two subspaces are nearly parallel.

To further understand the subtle variations in orientation, we performed PCA separately on the set of first principal vectors $\{v_{k1}\}_{k=0}^{24}$, the set of second principal vectors $\{v_{k2}\}_{k=0}^{24}$, and the set of third principal vectors $\{v_{k3}\}_{k=0}^{24}$ across all initial conditions. As shown in Figure 8 (dot product heatmaps) and Figure 9 (PCA of principal vectors), for the set of first principal vectors $\{v_{k1}\}$, the first PC explained 85.45% of their variance. For $\{v_{k2}\}$, the first PC explained 80.04%. Most notably, for $\{v_{k3}\}$, the first PC explained 97.67% of the variance. This indicates that the variations in the orientations of the principal axes of the 3D manifolds are themselves highly structured and change in a low-dimensional manner, effectively tracing out nearly one-dimensional paths in the space of orientation vectors as the initial condition index changes.

In summary, the 3D latent manifolds are not only translated versions of each other but also exhibit a very high degree of parallelism. The minor deviations in their orientations are systematic and follow a simple, low-dimensional pattern correlated with the initial condition index.

## 3.5 Relationship between per initial condition structures and global structure

Finally, we related the geometrically characterized per-IC manifolds to the overall structure of the global latent space. The global PCA identified a 6-dimensional subspace capturing 99.72% of the total variance (Figure 1). We projected the centered latent vectors $(L_k - C_k)$ for each initial condition $k$ onto this 6D global principal subspace. As shown in Figure 10, on average, 99.66% of each individual IC's intrinsic variance (the variance within its 3D manifold) was captured by this 6D global subspace, with a minimum capture of 99.24%. This confirms that the individual 3D manifolds are almost entirely embedded within the common, higher-dimensional subspace occupied by the entire dataset.

Furthermore, we projected the per-IC centroids $C_k$ onto the global principal components. This analysis, visualized in Figure 11 (2D projection) and Figure 12 (3D projection), showed that the trajectory of the centroids aligns strongly with the first global principal component (Global PC1). The initial condition index (0-24) maps almost linearly to the position along Global PC1. This demonstrates that the dominant mode of variation in the entire latent space (Global PC1) is directly associated with the primary way the initial conditions are encoded – as translations of the latent manifold along this direction.

These results highlight a hierarchical structure: a global 6D subspace accommodates all learned representations. Within this subspace, each specific initial condition selects a 3D affine manifold whose position is determined by a translation along a nearly 1D path strongly aligned with the global PC1. The orientation of this 3D manifold is remarkably consistent across ICs, with subtle, structured, low-dimensional variations.

## 3.6 Synthesis and interpretation

The collective findings from our geometric analysis provide a clear and compelling picture of how the PINN structures its latent space to represent solutions of the 2D Burger's equation across varying initial conditions. The latent space is not a complex, entangled high-dimensional mess but rather exhibits a highly organized geometric structure.

For a given initial condition, the network learns a representation that effectively lies on a 3-dimensional affine manifold within the 10-dimensional latent space. This intrinsic dimensionality is strikingly

consistent across all 25 tested initial conditions, as shown in Figure 3. The primary effect of changing the initial condition is not to drastically alter the structure or dimensionality of this manifold, but rather to translate it within the latent space. These translations occur along a well-defined, nearly one-dimensional path (Figures 5, 6), which is itself strongly aligned with the dominant direction of variation in the overall latent space (Figures 11, 12). Moreover, the orientation of these 3D manifolds is remarkably similar across different initial conditions, indicating they are nearly parallel (Figure 7). The subtle variations in their orientation are not random but follow a structured, low-dimensional pattern related to the initial condition (Figure 9).

This suggests that the PINN has learned a form of disentangled representation. The network appears to separate the influence of the initial condition from the intrinsic spatiotemporal evolution of the solution. The intrinsic dynamics for a fixed initial condition are encoded within the 3D structure of the manifold, while the specific initial condition primarily acts as a parameter that translates this fundamental 3D structure in the latent space. This organization is highly efficient; instead of learning 25 distinct, unrelated high-dimensional structures, the network leverages a common 3D "template" and uses a simple, low-dimensional transformation (translation and minor orientation adjustment) to adapt it for different initial conditions. This geometric simplicity in the latent space provides valuable insights into the network's internal encoding mechanisms, suggesting that the PINN captures the essential physics in a structured and interpretable manner, at least within this learned latent representation.

## 3.7   Limitations and future directions

While the findings reveal a surprisingly simple and structured latent space geometry, it is important to consider potential limitations and avenues for future research. Our analysis heavily relies on PCA, which is a linear technique. Although the high variance capture suggests that affine manifolds are good approximations, non-linear manifold learning techniques could potentially uncover finer, non-linear structures within the 3D manifolds or in the arrangement of centroids and orientations. The study was conducted for a fixed viscosity parameter; exploring how the latent space structure changes with varying viscosity would be a crucial extension, providing insights into how the PINN encodes physical parameters beyond initial conditions. A larger and more diverse set of initial conditions could further validate the observed low-dimensional nature of the centroid path and orientation variations, potentially revealing more complex patterns if the range of initial conditions were significantly expanded. Furthermore, correlating the specific characteristics of the initial conditions (e.g., amplitude, frequency content) with their positions along the centroid trajectory and their manifold orientations would provide deeper physical meaning to the learned latent structure. Finally, investigating whether similar structured latent spaces are learned by PINNs for other types of PDEs or with different network architectures is essential to assess the generalizability of these findings.

## 4   Conclusions

This study investigated the geometric structure of the 10-dimensional latent space generated by a Physics-Informed Neural Network (PINN) trained to solve the 2D Burger's equation across a set of 25 distinct initial conditions. Our goal was to understand how the PINN encodes the physical state of the system and how variations in the initial condition are reflected in the network's internal representation. We hypothesized that the latent space might exhibit a structured, potentially low-dimensional, organization related to the problem parameters.

To address this, we employed Principal Component Analysis (PCA) and subspace similarity measures to analyze the collections of latent vectors. We performed PCA on the entire dataset of latent vectors (global PCA), on the latent vectors for each individual initial condition (per-IC PCA), and on the centroids of the per-IC latent point clouds. Subspace similarity was used to compare the orientations of the principal subspaces identified by the per-IC PCA. The dataset comprised 10-dimensional latent vectors extracted from a pre-trained PINN solution for 25 initial conditions, sampled over a spatial and temporal grid.

Our analysis yielded several key findings regarding the geometric organization of the latent space. Globally, the latent vectors across all initial conditions occupy an effectively 6-dimensional subspace, capturing over 99% of the total variance, indicating an overall efficient representation. More specifi-

cally, for each individual initial condition, the set of latent vectors forms a distinct, approximately 3-dimensional affine manifold embedded within the 10-dimensional space. This intrinsic dimensionality and the distribution of variance along the principal components were remarkably consistent across all 25 initial conditions. Crucially, the primary effect of changing the initial condition is encoded as a translation of this consistent 3D manifold within the latent space. The centroids of these manifolds trace a nearly one-dimensional path, strongly aligned with the dominant global principal component, as the initial condition changes. Furthermore, the 3D manifolds for different initial conditions are remarkably parallel to each other, exhibiting an average subspace similarity exceeding 0.98, with only subtle, low-dimensional variations in their orientation.

From these results, we learned that the PINN develops a highly structured and geometrically simple representation of the Burger's equation solutions. Instead of learning entirely distinct high-dimensional representations for each initial condition, the network appears to learn a fundamental, low-dimensional (3D) structure representing the spatiotemporal evolution of the system for a fixed initial state. The specific initial condition then acts primarily as a parameter that translates this base structure along a specific direction in the latent space. This suggests a form of disentangled representation, where the network separates the influence of the initial condition (encoded as a translation) from the intrinsic dynamics (encoded within the 3D manifold structure). This geometric organization is highly efficient and offers valuable insights into the network's internal encoding mechanisms, suggesting that the PINN captures the essential physics in a structured and potentially interpretable manner within this latent space.

While our findings reveal a compelling geometric structure, it is important to acknowledge the limitations of relying primarily on linear techniques like PCA, which might miss finer non-linear structures. Future work could explore non-linear manifold learning techniques to further probe the geometry. Expanding the study to include variations in physical parameters, such as viscosity, and analyzing a larger, more diverse set of initial conditions would be crucial to assess the generalizability of these findings and potentially uncover more complex organizational principles. Correlating specific properties of the initial conditions with the latent space features (centroid position, manifold orientation) would provide deeper physical meaning. Finally, investigating whether similar structured latent spaces are learned by PINNs for other types of PDEs and with different network architectures is essential to determine the broader applicability of these observed geometric principles.

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

## A  Supplementary Figures

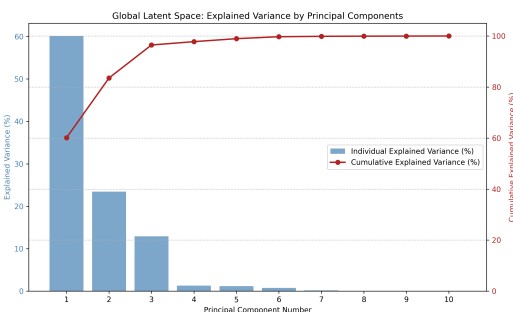

Figure 1: Scree plot showing the individual and cumulative explained variance from the global Principal Component Analysis of all latent vectors. The variance is highly concentrated in the first three components, which capture over 96% of the variance, revealing the low-dimensional structure of the global latent space.

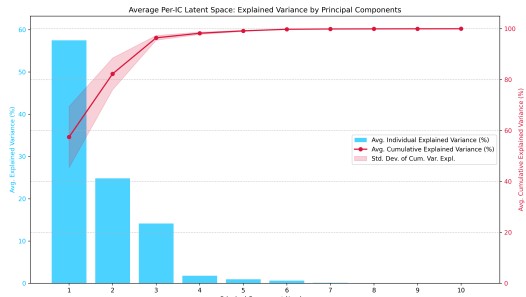

Figure 2: Average explained variance by principal components for the latent space of each initial condition, averaged across 25 initial conditions. Blue bars show the average individual explained variance per component; the red line shows the average cumulative explained variance with standard deviation (shaded). This analysis reveals that the latent representation for each initial condition is consistently low-dimensional, with the first three components capturing nearly 97.5% of the variance on average.

## B  Technical Appendices and Supplementary Material

Technical appendices with additional results, figures, graphs and proofs may be submitted with the paper submission before the full submission deadline, or as a separate PDF in the ZIP file below before the supplementary material deadline. There is no page limit for the technical appendices.

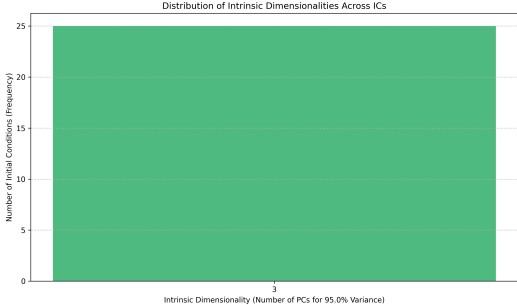

Figure 3: Distribution of the intrinsic dimensionality for the latent representations of each of the 25 initial conditions (ICs). Intrinsic dimensionality is defined as the minimum number of principal components required to capture over 95% of the variance for each IC's latent vectors. The plot shows that all 25 ICs result in latent manifolds with an intrinsic dimensionality of 3.

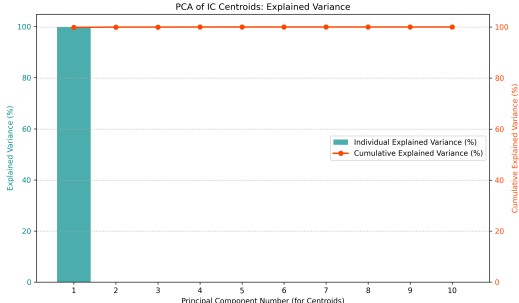

Figure 4: Scree plot showing the variance explained by principal components of the initial condition (IC) centroids. The first principal component captures over 99% of the variance, indicating that the centroids are arranged along an effectively one-dimensional structure in the latent space.

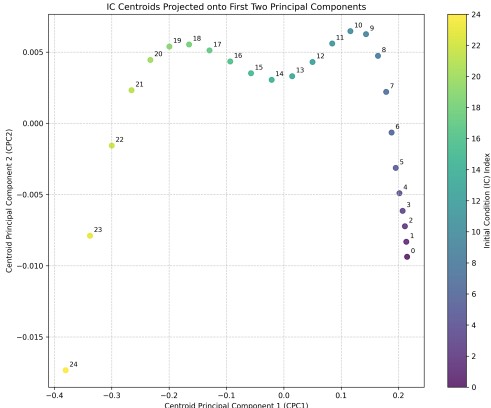

Figure 5: Initial condition (IC) manifold centroids projected onto their first two principal components (CPC1 and CPC2). Each point represents the centroid for a specific IC, labeled and colored by its index (0-24). The points form a clear, near-linear trajectory predominantly along CPC1, indicating that changing the IC primarily translates the corresponding latent manifold along a dominant direction.

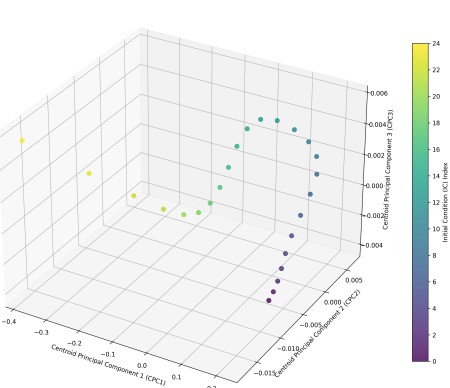

Figure 6: Three-dimensional scatter plot showing the projection of the 25 per-initial condition (IC) latent manifold centroids onto their first three principal components (CPC1, CPC2, and CPC3). Each point represents the centroid for a unique initial condition and is colored according to its corresponding IC index (0 to 24). The plot demonstrates that the centroids are arranged along a predominantly one-dimensional path, strongly aligned with CPC1, indicating that the primary effect of varying the initial condition is to translate the latent manifold along a specific direction.

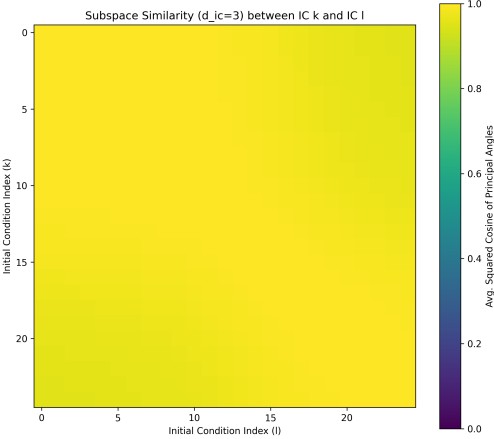

Figure 7: Subspace similarity between 3D latent manifolds for different initial conditions. The heatmap shows the average squared cosine of the principal angles between the subspaces spanned by the top three principal components for each pair of initial conditions ($IC_k$ and $IC_l$). High values (bright yellow) indicate strong alignment. The consistently high similarity across all pairs demonstrates that the 3D latent manifolds associated with different initial conditions are highly parallel.

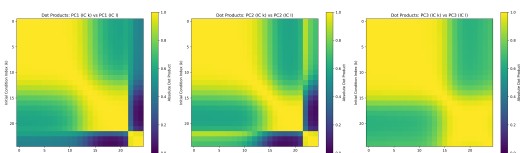

Figure 8: Heatmaps show the absolute dot product between corresponding principal vectors (PC1, PC2, PC3) from per-initial condition PCA for all pairs of initial conditions. High values (yellow) indicate strong alignment. The plots demonstrate substantial alignment across initial conditions, particularly for PC3, indicating that the 3D latent manifolds for different initial conditions are largely parallel.

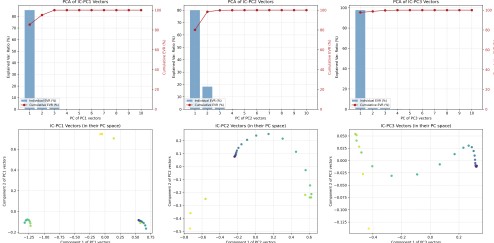

Figure 9: Principal Component Analysis (PCA) of the sets of per-initial condition (IC) principal vectors. Top row shows scree plots for the collection of first ($v_{k1}$), second ($v_{k2}$), and third ($v_{k3}$) per-IC principal vectors across all 25 ICs, indicating high variance capture by the first component in each set. Bottom row shows the 2D projection of these vector sets onto their respective first two principal components, colored by IC index, revealing a structured, low-dimensional variation in the orientation of the 3D per-IC manifolds.

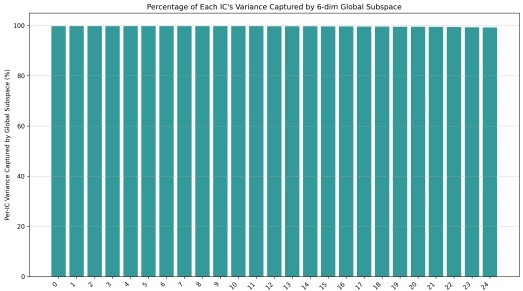

Figure 10: Percentage of the intrinsic variance for each initial condition (IC) latent manifold captured by the 6-dimensional global principal subspace. The consistently high values demonstrate that the individual 3D manifolds are effectively embedded within this common global subspace.

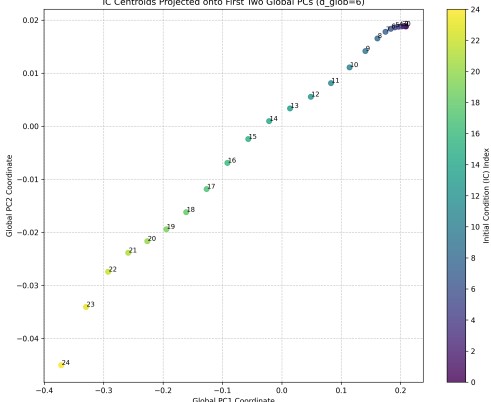

Figure 11: Projection of per-initial condition latent manifold centroids onto the first two global principal components. Each point, labeled and colored by initial condition index, reveals a near-linear arrangement predominantly along the first global component. This indicates that the PINN encodes variations due to initial conditions primarily by translating the corresponding latent manifolds along a structured, low-dimensional trajectory within the global latent space.

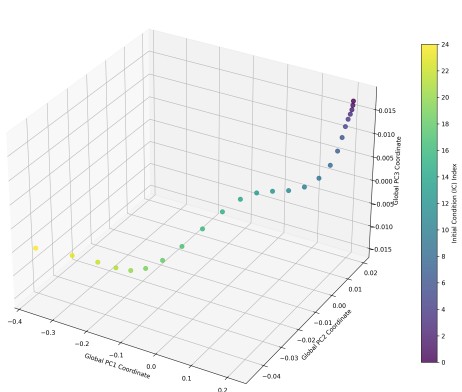

Figure 12: Centroids of the 25 per-initial condition (IC) latent manifolds projected onto the first three global principal components (PCs). Points are colored by IC index (0-24). The centroids form a near-linear path, primarily along Global PC1, indicating that different initial conditions primarily translate the latent manifolds along this dominant direction in the global latent space.

## Agents4Science AI Involvement Checklist

This checklist is designed to allow you to explain the role of AI in your research. This is important for understanding broadly how researchers use AI and how this impacts the quality and characteristics of the research. **Do not remove the checklist! Papers not including the checklist will be desk rejected.** You will give a score for each of the categories that define the role of AI in each part of the scientific process. The scores are as follows:

- **A. Human-generated**: Humans generated 95% or more of the research, with AI being of minimal involvement.

- **B. Mostly human, assisted by AI**: The research was a collaboration between humans and AI models, but humans produced the majority (>50%) of the research.

- **C. Mostly AI, assisted by human**: The research task was a collaboration between humans and AI models, but AI produced the majority (>50%) of the research.

- **D. AI-generated**: AI performed over 95% of the research. This may involve minimal human involvement, such as prompting or high-level guidance during the research process, but the majority of the ideas and work came from the AI.

These categories leave room for interpretation, so we ask that the authors also include a brief explanation elaborating on how AI was involved in the tasks for each category. Please keep your explanation to less than 150 words.

1. **Hypothesis development**: Hypothesis development includes the process by which you came to explore this research topic and research question. This can involve the background research performed by either researchers or by AI. This can also involve whether the idea was proposed by researchers or by AI.

   Answer: D

   Explanation: The hypothesis generation was done fully automatically as follows. Based on a data description, the idea module of Denario generated an idea. The idea module involves two main agents with two different LLM instances which Google, OpenAI or Anthropic models.

2. **Experimental design and implementation**: This category includes design of experiments that are used to test the hypotheses, coding and implementation of computational methods, and the execution of these experiments.

   Answer: D

Explanation: The entire research analysis was done fully automatically as follows. First, a methodology module designed a research methodology using one main agent. Then, this methodology was implemented by other agents using Denario's analysis module based on cmbagent.

3. **Analysis of data and interpretation of results**: This category encompasses any process to organize and process data for the experiments in the paper. It also includes interpretations of the results of the study.

   Answer: D

   Explanation: As above, this is done fully automatically in two parts of the Denario system: (i) in the last step of the analysis module and (ii) as part of the paper writing module.

4. **Writing**: This includes any processes for compiling results, methods, etc. into the final paper form. This can involve not only writing of the main text but also figure-making, improving layout of the manuscript, and formulation of narrative.

   Answer: D

   Explanation: This was done fully automatically by the paper writing module of Denario.

5. **Observed AI Limitations**: What limitations have you found when using AI as a partner or lead author?

   Description: As of now, we can not control the page limit.

