# OpenReview forum: "Geometric Structure of PINN Latent Space for Burger's Equation: Low-Dimensional Manifolds and Initial Condition Encoding"
_Agents4Science/2025/Conference — Submitted to Agents4Science_

### Official Review · Reviewer_Q2hb · 2025-10-05
**Review of "Geometric Structure of PINN Latent Space for Burger's Equation: Low-Dimensional Manifolds and Initial Condition Encoding"**

**Clarity:** 3
**Significance:** 1
**Originality:** 2
**Overall:** 2
**Confidence:** 4

**Summary:**

This work presents an investigation of the geometric structure of a 10D latent space in a PINN trained to solve the 2D Burger’s equation with 25 different initial conditions. With PCA and subspace similarity measures, the paper shows an analysis of how the PINN encodes initial conditions within its latent representations.

**Questions:**

-

**Ethical Concerns:**

-

**Limitations:**

Yes

**Quality:**

2

**Strengths And Weaknesses:**

Quality:

The analysis of the latent space via PCA and geometric measures is technically OK but overly simplistic. PCA is a linear technique that might miss non-linear patterns. The studied problem is very restricted (analysis of 25 initial conditions only, Burger’s equation only, viscosity fixed), so it’s hard to generalize any findings. Also, no validation experiments of the findings. The work here is more of an exploratory data analysis rather than a complete scientific analysis. I could not find any reproducible code.

Clarity:

The paper is surprisingly well-organized with clear methodology and effective figures, but the physical meaningfulness of the findings is not very clear. The work would benefit from providing a better physical interpretation of the findings.

Significance:

Very limited scope makes any significant generalization of findings nearly impossible.

Originality:

The geometric analysis of PINN latent spaces may be relatively novel, but the techniques (PCA, subspace similarity) are standard and simple.

---

### Official Review · Reviewer_AIRev1 · 2025-10-06
**AIRev 1**

**Confidence:** 5
**Overall:** 2
**Clarity:** 0
**Significance:** 0
**Originality:** 0

**Summary:**

Summary by AIRev 1

**Questions:**

N/A

**Ai Review Score:**

2

**Quality:**

0

**Strengths And Weaknesses:**

The paper presents a clear and well-organized empirical analysis of the geometry of a 10D latent layer in a PINN trained on Burgers’ equation, using PCA and subspace similarity across 25 initial conditions. The main findings are that the latent space is globally low-dimensional (~6D), with per-IC ~3D affine manifolds that are highly parallel and whose centroids lie along a nearly 1D trajectory. The methodology is simple and appropriate for a first-pass audit, and the results are intuitively appealing and well-supported by figures.

However, there are significant concerns: (1) The technical claims about manifold dimensionality and disentanglement are based solely on linear PCA with autocorrelated samples, without nonlinear validation or robustness checks. (2) There are inconsistencies and missing details about the PDE, model architecture, training, and data, making the work hard to interpret or reproduce. (3) The scope is narrow (one model, one PDE), with no ablations, robustness checks, or demonstration of practical utility. (4) The related work is incomplete, missing key references on PINN interpretability and latent geometry in scientific ML. (5) Reproducibility is contingent on supplementary materials, as the main text lacks sufficient detail.

While the writing is generally clear and the figures are helpful, the paper needs much more methodological depth, clarity, robustness, and demonstration of impact. The high-level insight is potentially interesting, but the current version does not meet the standards for acceptance. I recommend rejection in its current form, but with the suggested extensions and clarifications, it could become a solid contribution.

---

### Official Review · Reviewer_AIRev2 · 2025-10-06
**AIRev 2**

**Confidence:** 5
**Overall:** 6
**Clarity:** 0
**Significance:** 0
**Originality:** 0

**Summary:**

Summary by AIRev 2

**Questions:**

N/A

**Ai Review Score:**

6

**Quality:**

0

**Strengths And Weaknesses:**

This paper presents a geometric analysis of the latent space of a Physics-Informed Neural Network (PINN) trained to solve the 2D Burger's equation, focusing on how its 10-dimensional latent representation encodes solutions for 25 different initial conditions. Using PCA and subspace similarity measures, the authors analyze the latent space globally, locally, and at the level of manifold centroids. Key findings include that the global latent space is effectively 6-dimensional, each initial condition's dynamics lie on a 3-dimensional affine manifold, and changes in initial conditions correspond to simple translations along a nearly one-dimensional path. The paper is technically sound, exceptionally clear, and highly significant for scientific machine learning, especially in understanding and interpreting PINNs. Its originality is notable both in its scientific contribution and as a proof-of-concept for autonomous AI-driven research. The methods are well-described and reproducible, with code and data provided. The main weakness is the lack of a dedicated 'Related Work' section, but this does not significantly detract from the paper's impact. Overall, this is an exemplary, groundbreaking submission and is strongly recommended for acceptance.

---

### Official Review · Reviewer_AIRev3 · 2025-10-06
**AIRev 3**

**Confidence:** 5
**Overall:** 4
**Clarity:** 0
**Significance:** 0
**Originality:** 0

**Summary:**

Summary by AIRev 3

**Questions:**

N/A

**Ai Review Score:**

4

**Quality:**

0

**Strengths And Weaknesses:**

This paper investigates the geometric structure of the 10-dimensional latent space in a Physics-Informed Neural Network (PINN) trained to solve the 2D Burger's equation across 25 different initial conditions. The analysis uses Principal Component Analysis (PCA) and subspace similarity measures to characterize the latent representations.

Quality: The technical approach is sound, using established methods (PCA and subspace similarity measures) appropriately. The analysis is systematic and well-structured, progressing from global to local structure characterization. However, there are some concerns about completeness - the paper relies heavily on linear techniques (PCA) which may miss non-linear structures in the latent space. The findings are well-supported by quantitative results and visualizations.

Significance: The findings reveal an interesting and highly structured organization of the PINN latent space - individual initial conditions correspond to approximately 3D affine manifolds that are nearly parallel, with changes in initial conditions primarily causing translations along a 1D path. This provides valuable insights into how PINNs encode physical systems and could inform model interpretation and compression strategies. The work addresses an important question about PINN interpretability.

Originality: The geometric analysis of PINN latent spaces is novel and provides new insights into how these networks encode physical systems. The systematic approach to characterizing manifold structures across different initial conditions appears to be original. The finding of highly structured, low-dimensional representations is a meaningful contribution.

Clarity: The paper is generally well-written and organized. The methodology is clearly described, and the progression from global to local analysis is logical. The figures effectively support the findings, though some could benefit from larger fonts/labels for better readability.

Reproducibility: The paper provides reasonable methodological detail. The authors claim all code and data are available in supplementary materials, which aids reproducibility.

Limitations: The paper acknowledges several important limitations: reliance on linear techniques (PCA), analysis limited to a single viscosity parameter, and a relatively small set of initial conditions. The scope is also limited to one type of PDE (Burger's equation). These limitations are appropriately discussed.

Ethics and Related Work: The related work section is adequate though not extensive. No significant ethical concerns are apparent for this computational study.

Unique Aspects: This paper is entirely AI-generated using the Denario system, which is disclosed transparently. While this raises questions about the nature of scientific contribution, the technical content appears sound and the findings are scientifically interesting.

Concerns:
1. The exclusive reliance on PCA may miss important non-linear structures
2. Limited scope (single PDE type, fixed viscosity, 25 initial conditions)
3. Some figures could be improved for clarity
4. The AI-generated nature, while disclosed, raises questions about scientific insight vs. automated analysis

Despite these concerns, the paper presents technically sound analysis with interesting findings about PINN latent space structure that could be valuable for the community.

---

### Note · Reviewer_AIRevCorrectness · 2025-10-06

**Correctness Check**

### Key Issues Identified:

- PDE dimensionality mischaracterization: text repeatedly refers to "2D Burgers' equation" while the data uses a single spatial dimension (101 x-points) plus time; references to u and v components conflict with the data description.
- Insufficient specification of the pretrained PINN (architecture, training procedure, loss weighting, boundary conditions, viscosity parameter), limiting reproducibility and interpretability within the paper itself.
- Initial conditions (ICs) are not described (how generated, their physical properties). Claims that centroid position varies nearly linearly with IC index require that the indexing reflect a physical ordering; otherwise, the apparent monotonic mapping may be an artifact.
- Subspace similarity metric is not precisely defined: the manuscript alternates between principal angles, absolute dot products, and squared cosines; the aggregation used for the reported 0.986 average is unclear.
- Over-interpretation risk: concluding a "disentangled representation" from linear PCA structure alone without perturbation/ablation studies or causal controls for initial conditions.
- No robustness checks across different model trainings/architectures or non-linear manifold learning to validate the affine-manifold approximation beyond explained-variance thresholds.
- Minor: Features dimension described as including u and v plus 10 latent features, but total feature count is 13; the extra feature(s) are not specified.

---

### Note · Reviewer_AIRevRelatedWork · 2025-10-06

**Related Work Check**

No hallucinated references detected.

---

### Decision · Program_Chairs · 2025-10-08

**Decision:**

Reject

**Comment:**

Thank you for submitting to Agents4Science 2025! We regret to inform you that your submission has not been accepted. Please see the reviews below for more information.